# Current Applications and Future Directions of Circulating Tumor Cells in Colorectal Cancer Recurrence

**DOI:** 10.3390/cancers16132316

**Published:** 2024-06-24

**Authors:** Kun-Yu Tsai, Po-Shuan Huang, Po-Yu Chu, Thi Ngoc Anh Nguyen, Hsin-Yuan Hung, Chia-Hsun Hsieh, Min-Hsien Wu

**Affiliations:** 1Division of Colon and Rectal Surgery, New Taipei Municipal TuCheng Hospital, New Taipei City 23652, Taiwan; edward31705@cgmh.org.tw (K.-Y.T.); hsinyuan@cgmh.org.tw (H.-Y.H.); 2Graduate Institute of Biomedical Engineering, Chang Gung University, Taoyuan City 33302, Taiwan; leo_6813@cgu.edu.tw (P.-S.H.); d000018394@cgu.edu.tw (P.-Y.C.); d1131003@cgu.edu.tw (T.N.A.N.); 3College of Medicine, Chang Gung University, Taoyuan City 33302, Taiwan; wisdom5000@cgmh.org.tw; 4Division of Hematology and Oncology, Department of Internal Medicine, Chang Gung Memorial Hospital at Linkou, Taoyuan City 33302, Taiwan; 5Division of Hematology and Oncology, Department of Internal Medicine, New Taipei Municipal Hospital, New Taipei City 23652, Taiwan; 6Department of Biomedical Engineering, Chang Gung University, Taoyuan City 33302, Taiwan

**Keywords:** circulating tumor cells, CTCs, CTC subtypes, colorectal cancer recurrence, adjuvant chemotherapy, neoadjuvant chemoradiotherapy, clinical application, multiple parameters

## Abstract

**Simple Summary:**

The ability to predict or detect colorectal cancer (CRC) recurrence early after surgery enables physicians to apply appropriate treatment plans and different follow-up strategies to improve patient survival. A certain portion of CRCs will recur, and current surveillance tools have limitations in the precise and early detection of cancer relapse. Circulating tumor cells (CTCs), cancer cells that are disconnected from the primary tumor and enter the bloodstream, can provide real-time information on disease status. CTCs might become markers not only for predicting CRC recurrence but also for guiding therapy for stage II CRC and monitoring disease relapse in locally advanced rectal cancer patients receiving neoadjuvant therapy. Using CTC subtypes and CTCs combined with clinicopathological factors is an even more powerful way to predict CRC recurrence.

**Abstract:**

The ability to predict or detect colorectal cancer (CRC) recurrence early after surgery enables physicians to apply appropriate treatment plans and different follow-up strategies to improve patient survival. Overall, 30–50% of CRC patients experience cancer recurrence after radical surgery, but current surveillance tools have limitations in the precise and early detection of cancer recurrence. Circulating tumor cells (CTCs) are cancer cells that detach from the primary tumor and enter the bloodstream. These can provide real-time information on disease status. CTCs might become novel markers for predicting CRC recurrence and, more importantly, for making decisions about additional adjuvant chemotherapy. In this review, the clinical application of CTCs as a therapeutic marker for stage II CRC is described. It then discusses the utility of CTCs for monitoring cancer recurrence in advanced rectal cancer patients who undergo neoadjuvant chemoradiotherapy. Finally, it discusses the roles of CTC subtypes and CTCs combined with clinicopathological factors in establishing a multimarker model for predicting CRC recurrence.

## 1. Prevention and Detection of Colorectal Cancer Recurrence

Colorectal cancer (CRC) is one of the most diagnosed cancers and the second leading cause of cancer deaths globally [1], making it a considerable health burden worldwide [2]. Around 80% of newly diagnosed CRC patients present with nonmetastatic disease [3] in which cancer cells do not spread from the primary tumor to other organs. For most nonmetastatic CRC patients, complete removal of the malignant tumor by radical surgery is the gold-standard treatment [4,5]. With the combination of more advanced screening tools, enhanced chemotherapy drugs and radiotherapy equipment, and surgical techniques, the risk of postoperative recurrence incidence has decreased yearly [6]. Still, some 30 to 50% of CRC patients still experience disease recurrence after radical surgery [7,8]. More concerning, around 25% of patients with theoretically curable stage I and II disease have a recurrence [9]. Cancer recurrence has a direct adverse impact on patient prognosis. For patients without recurrence at 1 year after surgery, the 5-year CRC related death rate was only 3.8%, but it increased to 33.6% for those patients with recurrence [10]. Therefore, applying adjuvant chemotherapy to certain patients at high risk for recurrence [11,12,13] and using different monitoring strategies to detect recurrence are two crucial steps to prolong patient survival after surgery [4,5]. 

In clinical practice, adjuvant chemotherapy aims to eradicate residual tumor cells that might be undetectable by the current method to reduce the recurrence rate or delay the time to recurrence after curative surgery [14]. The choice of adjuvant chemotherapy is based on the risk of recurrence, which usually depends on pathological tumor staging [4,5]. The recurrence rate is 3–5% in stage I CRC patients, 10–17% in stage II CRC patients and 31–40% in stage III CRC patients during follow-up [14,15]. Moreover, stage III CRC patients experience disease relapse sooner than the early-stage patients. The median durations of recurrence after surgery for stage I, II and III CRC patients were 22.6 months, 18.2 months and 15.9 months, respectively [6]. Therefore, several similar CRC guidelines strongly suggest that stage III CRC patients should receive adjuvant chemotherapy [11,12], but such treatment is unnecessary for stage I patients. For stage II CRC, adjuvant chemotherapy is not routinely recommended and is given only for patients at high risk of recurrence [11,13] (Figure 1). For these patients, multiple clinical and histopathological factors, including advanced tumor stage, tumor perforation, intestinal obstruction, poorly differentiated histology, lymphovascular invasion, or perineural invasion, have been evaluated as risk factors for recurrence. For instance, a study found that the 5-year disease-free survival (DFS) rate was 87.3% for stage II CRC patients with only one risk factor, but it dropped to 74.8% for patients with two or more risk factors [16]. Gertler et al. also found that the 5-year recurrence-free survival (RFS) of patients with stage II colon cancer decreased from 91% with no risk factor to 75% with three risk factors [17]. Nevertheless, current evidence has not identified any of the regularly assessed parameters as individually powerful enough for risk stratification in high-risk stage II CRC [18]. Research to better identify risk factors for those patients who may benefit from adjuvant chemotherapy is needed.

Although colon and rectal cancers are both called ‘colorectal cancer’ in all fields of research and clinical practice, obvious differences exist between them in many ways, including anatomic location, physical function, and surgical procedures [19]. Because of the anatomy of the rectum and its main function in the storage of feces and defecation [20], surgical operations for rectal cancer patients face more challenges and higher risks of complications and irreversible damage, such as anal incontinence and malfunctions of pelvic organs, than colon surgery. Therefore, some treatment strategies, known as neoadjuvant therapies, will shift parts of the adjuvant treatment before surgery to combat this more complicated disease. Neoadjuvant therapy can be radiotherapy, chemotherapy, or both before radical resection in patients with locally advanced rectal cancer (LARC). In the neoadjuvant setting, a portion of patients could benefit from tumor shrinkage in advance of curative surgery [21] and earlier treatment of micrometastatic disease to improve their DFS [22] (Figure 1). Nevertheless, because of the tumor heterogeneity [23] (the differences between the same types of cancer in different patients and the differences between cancer cells within a single tumor), a certain amount of rectal cancer will not respond completely to the intensified neoadjuvant treatment [24] and may delay the opportunity for surgical resection, leading to tumor development during this period [25]. Only some predictive markers (e.g., tumor stage, size, differentiation, and tumor budding, specific genes, microRNAs, and proteins) are used in the clinical practice to predict the response to neoadjuvant therapy before treatment [26], so a novel predicting marker(s) is warranted to address this issue.

In addition to the issues related to adjuvant chemotherapy described above, to detect cancer recurrence early, patients should receive a cancer surveillance protocol after surgery to monitor recurrence [4,5]. This is because patients might benefit from more surgical resection options, chemotherapeutic treatments, or even targeted therapies if recurrence is detected early and precisely, which leads to improved treatment success and therefore improved long-term survival [7]. Generally, the most common form of recurrence of CRC is distant metastasis (78%), followed by local recurrence (18%) and both (4%) [27]. The most common site of CRC recurrence is the liver, followed by the lungs [8], while rectal cancer patients more often experience local regional recurrence and isolated lung metastasis [28] because of the specific anatomy of venous drainage. Based on CRC recurrence-related characteristics, current surveillance guidelines for CRC patients recommend combining clinical examinations (history and physical examination), laboratory tests (serial carcinoembryonic antigen (CEA) level), colonoscopy, and radiological examinations (usually chest/abdominal/pelvic computed tomography (CT)), depending on the patient’s tumor stage. For example, according to the National Comprehensive Cancer Network (NCCN) guideline [4,5], colonoscopy is the only examination and is suggested for stage I colorectal cancer patients 1 year after surgery. For patients with stage II–III disease, a 3- to 6-monthly history, physical examination, and CEA measurements are advised for the first 2 years, followed by every 6 months for a total of 5 years, and a 6- to 12-monthly CT scan is recommended for a total of 5 years. Colonoscopy is performed 1 year after surgery, but for patients who undergo preoperative incomplete colonoscopy due to obstructive disease, colonoscopy must be completed 3–6 months postoperatively [29]. Even though these recommendations have been adopted by surgeons and oncologists and are routinely applied in clinical practice, current surveillance tools have certain limitations in clinical practice (Table 1).

### 1.1. History Taking and Physical Examinations

History taking and physical examination are recommended as the first investigations for all CRC patients after surgery [4,5,30]. Nonetheless, because of the lack of specific symptoms when early recurrence occur, it is often difficult to confirm the asymptomatic disease relapse by physical examination alone. Fewer than 7% of patients with symptoms caused by recurrence have resectable disease [31], implying that physical examination is often delayed and has lost its importance for early detection. 

### 1.2. Carcinoembryonic Antigen (CEA)

CEA is the most commonly used and cost-effective blood tumor marker for CRC recurrence and is recommended for every surveillance protocol [4,5,30]. This specific antigen of the human digestive system can be used as a continuous monitoring tool in recurrence monitoring [32]. CEA is a simple, safe, and minimally invasive marker with low risk, and patients do not need any special preparation before examination, so it can be measured often in clinical practice. Although the specificity of CEA for detecting recurrence after curative surgery is greater than 80%, its low sensitivity (50 to 80% depending on the cut-off value [33]) makes it difficult to detect recurrence alone [34]. Moreover, many factors have been reported to cause nonspecific abnormal CEA elevation, including cigarette smoking, inflammatory bowel disease, and even lung disease and hypothyroidism [35]. Such false-positive increases in CEA may occur in up to 20% of recurrence-free CRC patients after surgery [36]. 

### 1.3. Colonoscopy

Colonoscopy is an imaging-based invasive procedure in which a flexible fiber-optic instrument is inserted through the anus to examine the colon and rectum [37]. Colonoscopy is essential for detecting anastomotic (a connection between two bowel lumens via surgery) recurrence or second primary CRC [38]. This procedure does have its disadvantages, including the inconvenience of bowel preparations before examination and discomfort during invasive procedures, causing patient anxiety and rare but severe complications such as colon perforation and bleeding. Moreover, colonoscopy cannot easily detect extraluminal recurrence, one type of local recurrence in the presacral area, which occurs more often (15–30%) after rectal cancer resection [39,40]. 

### 1.4. Computed Tomography (CT)

For radiologic surveys, CT scanning is the most common examination in CRC surveillance. It produces consecutive images of the inside of the whole body and provides an effective means for detecting new lesions, especially in the liver or chest [41]. However, it is difficult to distinguish postoperative changes and recurrence when CT is used alone, especially in the pelvic area [9]. Its accuracy is relatively lower for detecting extrahepatic metastases and local recurrence [42]. CT involves much higher doses of radiation than other imaging tools and is not recommended for high-frequency use [43]. 

### 1.5. Positron Emission Tomography (PET)

PET is another radiological tool that uses small amounts of radioactive materials called radiopharmaceuticals to evaluate the metabolism of a particular organ or tissue [44]. Cancer cells exhibit bright spots on PET scans because they have a higher metabolic rate than nonmalignant cells [45]. Compared to CT and CEA, it is very accurate for the detection of local and/or distant recurrent disease in CRC patients with a high suspicion of recurrent disease [42,46]. PET scanning is also a method favored by many physicians because of its high accuracy in detection CRC recurrence, especially extra-hepatic lesions [41], but the examination is extremely expensive and highly demanding of physicians, making it impractical for routine surveillance use. According to NCCN guidelines [4,5], PET should be considered when serial elevation of CEA is noted but with a negative CT finding. 

Briefly, even though continuous monitoring and real-time detection of CRC recurrence and metastasis after surgery are important, these methods are still difficult to implement because of the complicated and unclear mechanism of CRC recurrence [47]. As mentioned above, more than 30 to 50% of CRC patients recur even after proper therapy and radical resection during follow up. Many patients already have micrometastases at the time of the operation that are not removed by surgery [48] and require systemic therapy. Many past studies on the ability to target cancer metastasis found that primary cancer cells spread to other organs via blood or lymphatics circulation; these cells are named circulating tumor cells (CTCs) [49]. Such cancer cells may be present in the bloodstream in extremely low numbers and would hardly be detected by conventional methods [50]. As a result, detection of CTCs in the blood has the potential to be a novel examination method and may have important prognostic and therapeutic implications in the clinic to assist physicians in detecting CRC recurrence. In this review, we first introduce the clinical application of CTCs as a prognostic factor for CRC recurrence and their potential role in therapeutic guidance for adjuvant chemotherapy. Next, we discuss the utility of CTCs for monitoring cancer recurrence in advanced rectal cancer patients who undergo neoadjuvant chemoradiotherapy. We also discuss the value of advanced CTC subtypes and the combined use of CTCs and other routine clinicopathological factors to provide CRC patients with more appropriate and effective strategies for CRC recurrence.

## 2. Characteristics of CTCs and the Prognostic Value of CTCs in CRC Recurrence

### 2.1. Circulating Tumor Cells

CTCs are rare cancer cells that detach from primary tumors or metastatic tumors and then circulate in the bloodstream [51]. It is believed that when CTCs are trapped in the capillaries of organs or tissue [52], the cancer will colonize and metastasize [53]. Although CTCs were first detected in patients with CRC more than 70 years ago [54], the significance and application of CTCs in CRC patients who undergo surgery began to increase 20 years ago thanks to developments in molecular technology. The characteristics of CTCs disseminated in the blood allow them to be easily collected by “liquid biopsy” from peripheral venous vessels [55]. This is a safe and minimally invasive procedure compared to radiological scans and colonoscopy because it can be performed often and provides real-time monitoring of cancer recurrence.

### 2.2. Presence of CTCs and Postoperative CRC Recurrence

Earlier studies have shown that the presence of CTCs in patient blood might be a potential prognostic marker for predicting CRC recurrence after surgery [56]. In the early, developmental stage of CTC research, most CTC detection methods included reverse-transcription polymerase chain reaction (RT-PCR), which uses specific mRNA expression in CTCs to identify these cells. For instance, Ito et al. reported that the expression of CEA mRNA in the CTCs of 99 nonmetastatic CRC patients after surgery was associated with shorter DFS than the absence of CTCs [50]. Moreover, another study revealed that the CTC-related mRNA detection 24 h after primary tumor surgery is an independent risk factor for cancer recurrences [57]. Several studies applying multivariate statistical analysis came to similar conclusions [58,59,60,61,62,63]. 

Additionally, CTCs might be a stronger indicator of CRC recurrence than the current blood marker CEA. For instance, Wang et al. published a pivotal study to detect CTCs in CRC patients who underwent radical surgery, but their perioperative serum CEA levels were normal [64]. The results showed that positive CTC results were correlated with clinicopathologic characteristics, including depth of tumor invasion, lymph node metastasis, and postoperative recurrence, which indicates that CTCs can be an independent prognostic factor for CRC patients with low or normal CEA. The authors also found that the interval at which positive CTCs were detected was 6 months (range 3 to 8 months) earlier than that at which elevated CEA was detected. Likewise, in another cohort, elevated serum CEA and positive CTC results were the only two risk factors for postoperative recurrence, with odds ratios of 4.13 and 66.67, respectively [65]. Positive CTCs had higher sensitivity (87.5%) and specificity (92%) in predicting postoperative recurrence than CEA (60.4% and 83.2%, respectively), implying that the presence of CTCs was more powerful for predicting CRC recurrence than elevated CEA. Moreover, CTC positivity indicated recurrence around 8 months earlier than elevated CEA (median 10.7 vs. 2.8 months). These two studies demonstrate the potential role of CTCs in facilitating early postoperative CRC recurrence, which may improve the problem of the low detection sensitivity of CEA for CRC recurrence. In particular, CTCs, like CEA, which is a marker from blood tests, can be repeatedly checked at outpatient clinic follow-ups without any preparation and can be used for dynamic disease monitoring. 

Furthermore, as technology advances, some innovative CTC detection techniques have been proposed to simplify CTC detection operations (e.g., automated CTC detection equipment) or improve CTC detection performance. CellSearch^®^ (San Diego, CA, USA) is the first and only Food and Drug Administration (FDA)-approved system for the isolation, identification, and counting of CTCs based on multimarker staining (i.e., EpCAM+, CK+, CD45−, and DAPI+) [66]. It allows for the detection and enumeration of CTCs in a standardized fashion and has also been applied in studies of CTC detection and CRC recurrence. For instance, Bork et al. studied a prospective cohort of 239 stage I–III CRC patients [67] and reported that CTC positivity was strongly associated with poor overall survival (OS). Van Dalum et al. also analyzed the relationship between CTC detection by the CellSearch^®^ system and postoperative CRC recurrence [68]. They found that not only did patients with positive CTCs before surgery have significantly shorter RFS and cancer-related survival, but the presence of CTCs 2–3 years after surgery was correlated with patient survival. In addition to the CellSearch^®^ system, Wang et al. used immunofluorescence in situ hybridization (imFISH) staining to detect CTCs in 130 stage II–III CRC patients [69]. Their multivariable analysis found that postoperative CTCs were correlated with poor RFS. Yang et al. used the ISET (isolation by size of epithelial tumor cells) device CTCBIOPSY^®^ (Wuhan, China) to detect CTCs in 138 CRC patients [70]. The ISET device isolates epithelial tumor cells by their size [71], as a tumor marker-independent technology to detect CTCs. Patients with post-operative CTC positivity had a lower 3-year RFS than the post-operatively CTC-negative patients. The patients with persistent CTC positivity had a lower 3-year RFS than those patients positive for CTCs preoperatively but negative after surgery. Postoperative CTC positivity was independently associated with lower 3-year RFS. Therefore, based on similar clinical prognosis prediction based on the different CTC detection techniques described above, the presence of CTCs in postoperative CRC patients mostly suggests a poorer prognosis and higher recurrence risk. 

In addition to the abovementioned studies showing the presence of CTCs at a single point perioperatively (mostly 7 days after surgery), some studies have also explored the clinical value of the persistent presence of CTCs in predicting CRC recurrence. The persistent presence of CTCs is defined by CTC detection at two or more times; exactly when depends on the study design. The first CTC test is usually performed before surgery as a baseline, and the second CTC test is usually performed after surgery or after adjuvant chemotherapy. For example, one study demonstrated that the persistent presence of CTCs (1 day before surgery and 1 week after surgery) was an independent risk factor predicting CRC recurrence [72]. A retrospective study found that persistent postoperative CTCs (1 and 4 weeks after surgery) were a prognostic factor for early tumor recurrence (within 1 year after radical resection) [73]. Another study analyzed the correlation between persistent CTCs and the outcome of stage III CRC patients after adjuvant mFOLFOX (oxaliplatin and 5-FU) chemotherapy [74] and indicated that the persistent presence of postchemotherapy CTCs was an independent risk factor for cancer recurrence. Additionally, in terms of the accuracy of detecting CRC recurrence, the predictive performance of the persistent presence of postchemotherapy CTCs was greater than that of postchemotherapy CEA levels. Taken together, these studies demonstrate that dynamic CTC trends play an important role in CRC recurrence monitoring. In particular, the presence of CTCs after treatment (whether surgery or chemotherapy) implies that residual disease exists and that patients have a greater risk of poor outcomes (e.g., cancer recurrence). CTC detection at the end of every course of therapy may help physicians choose treatment plans and surveillance strategies in the future. 

In summary, CTC detection in the peripheral blood has the potential to become a prognostic factor for CRC recurrences [56,75,76] and disease progression [77]. The timing of blood tests can be applied at multiple points, including the time after surgery, since consecutive persisting CTCs or postoperative detection of CTCs implies a higher risk of cancer relapse.

## 3. Novel Utility of CTCs in Stage II CRC Patients Who Need to Receive Adjuvant Chemotherapy and in Advanced Rectal Cancer Patients Who Have Undergone Neoadjuvant Therapy

Recent studies have shown that the role of CTCs in CRC recurrence not only serves as a prognostic indicator but also plays an important role in guiding adjuvant chemotherapy in stage II CRC patients and monitoring treatment response and recurrence in advanced rectal cancer patients who undergo neoadjuvant therapy first. This section will discuss the novel utility of CTCs in these two specific clinical scenarios.

### 3.1. Clinical Application of CTCs as A Therapeutic Marker for Stage II CRC Patients Receiving Adjuvant Chemotherapy

As mentioned above, the purpose of adjuvant chemotherapy is to eliminate residual cancer cells after curative surgery to reduce the recurrence risk and postpone the time to recurrence [14]. Nevertheless, chemotherapy can lead to certain adverse events (e.g., diarrhea, nausea, vomiting, mouth pain, fatigue, appetite loss), impaired social life, inconvenience, and a financial burden. Therefore, due to the relatively low recurrence risk of stage II CRC patients, adjuvant chemotherapy is not routinely suggested by current guidelines [4,5]. However, stage II CRC patients with a high risk of recurrence may benefit from receiving adjuvant chemotherapy [78]. Although several clinicopathological parameters, including tumor staging, histopathology differentiation, and the number of lymph nodes removed, are prognostic factors that can be easily investigated in clinical practice, none of them is individually powerful enough for risk stratification in high-risk stage II CRC [18]. Some 10–17% of stage II CRC patients still experience cancer recurrence under current risk-assessment strategies [14], suggesting that risk evaluation via current parameters is insufficient. Therefore, in clinical practice, a novel indicator is needed to evaluate and distinguish stage II CRC patients who need to receive adjuvant chemotherapy. 

As described above, CTCs are a potential prognostic factor for evaluating nonmetastatic (stage I–III) CRC recurrence, but studies analyzing their utility in only stage II CRC patients are limited (Table 2). For instance, Koch et al. conducted a pilot study to analyze the presence of CTCs by RT-PCR in stage II CRC and its correlation with survival [58]. Cancer cells detected in blood 24 h after surgery were an independent prognostic factor for CRC recurrence. Another retrospective study analyzed the impact of the presence of CTCs on stage II CRC patient recurrence over a median follow-up of 40 months [79]. The presence of CTCs at one week after surgery in patients with colon or rectal cancer was strongly correlated with a significantly higher recurrence rate. More importantly, the presence of CTCs indicated postoperative recurrence 7 months (range 4 to 10 months) earlier than that indicated by the current clinical detection tool (CEA). Another study by Yu et al. applied the same study design but with immuno-fluorescence in situ hybridization (imFISH) staining in 73 stage II CRC patients [80] and found a significant difference in RFS between the CTC-positive group and CTC-negative patients (28.7 months versus 34.0 months). 

Because the presence of CTCs indicates a worse prognosis in patients with stage II CRC, some studies have investigated the potential of CTCs as a therapeutic marker in patients with stage II CRC (Table 2). Recently, one study demonstrated that the use of CTCs could guide the choice of adjuvant chemotherapy [81]. The study found that CTC-positive patients (CTCs ≥ 3) who did not receive adjuvant chemotherapy had shorter OS than CTC-negative patients who did not receive chemotherapy (68.1 months versus 82.0 months). Chen et al. conducted a larger population study involving 277 individuals with stage II CRC and obtained more promising results [82]. They used EpCAM-independent subtraction enrichment with immunostaining-fluorescence in situ hybridization (SE-iFISH) to isolate and count CTCs. In their cohort, patients with a preoperative CTC count ≥4 had a significantly higher recurrence risk than those with a preoperative CTC count <4. Moreover, if their CTC count was ≥4, patients showed a survival benefit if they received postoperative chemotherapy. More importantly, if a patient’s CTC count was ≥4 at more than three consecutive measurements (between 2 and 6 months), the recurrence rate was 100% in this cohort. This study suggested that the persistent presence of CTCs in the blood is a risk factor for CRC recurrence in these special groups, and the possible role of the dynamic monitoring of CTCs is essential in CRC surveillance. Taken together, the detection of CTCs in stage II CRC patients suggested the clinical utility of these cells in intense follow-up strategies and decision-making regarding adjuvant chemotherapy, which can be used to evaluate and distinguish stage II CRC patients who need to receive adjuvant chemotherapy.

### 3.2. The Clinical Value of CTCs in Rectal Cancer Patients Who Undergo Neoadjuvant Chemotherapy

Among clinical cases, rectal cancer accounts for around 35% of the CRC population, but the treatment strategy is more complex due to its specific anatomy [22] compared to that of colon cancer. Among the previously mentioned strategies for treating LARC, the most widely adopted strategy is neoadjuvant chemoradiotherapy (NCRT), followed by surgery and adjuvant chemotherapy. The purpose of neoadjuvant therapy is to shift part of the adjuvant treatment to before surgery. By following this approach, patients may benefit from better compliance than they would with postoperative chemotherapy and earlier treatment of micrometastatic disease to improve disease-free survival [22]. Around 10–30% of patients benefit from pathological complete remission (pCR) after NCRT, in which no residual tumor is detected in the specimen, which may facilitate the avoidance of surgical resection [83]. However, because of tumor heterogenicity not only between different patients with the same disease but also between cancer cells derived from the same tumor [23], only a few predictive markers have been applied in clinical practice to predict the response to neoadjuvant therapy beforehand [26]. The presence of CTCs or persistent CTCs detected after treatment (surgery) is a poor prognostic factor for CRC patients. Therefore, monitoring CTCs might have potential in assessing the therapeutic response of rectal cancer patients who undergo NCRT and predicting cancer recurrence in this specific group [84].

#### 3.2.1. Predicting Response to Neoadjuvant Chemoradiotherapy in Rectal Cancer Patients

Although neoadjuvant therapy is gaining popularity in treating LARC, a certain portion of rectal cancer patients do not respond well to intensified neoadjuvant treatment. Based on the concepts of personalized medicine, to predict treatment response, it is essential to choose appropriate treatment plans for every patient individually. Accumulating evidence indicates the clinical utility of CTCs in predicting the response to neoadjuvant chemotherapy in patients with many types of malignancies, including breast [85], esophageal [86], and rectal cancer patients [87]. Therefore, examining the waxing and waning of CTCs before and after treatment might help physicians evaluate therapeutic responses. For instance, one study used the detection of cytokeratin 20 (CK20) and CEA levels in patients positive for CTCs to examine their role in the therapeutic response of LARC patients to NCRT [88]. The results revealed a significant decrease in CTC positivity in responders during treatment compared to nonresponders. To objectively assess the NCRT response via tumor regression grade (TRG) [89], Sun et al. analyzed the association between CTC count and the tumor response to neoadjuvant chemotherapy via the CellSearch^®^ system [90]. A total of 115 LARC patients who received NCRT followed by radical surgery after 6–8 weeks were enrolled. According to the TRG, which is defined as the percentage of tumor cell reduction after treatment, patients were divided into responders (TRG 3–4) and nonresponders (TRG 0–2). Post-NCRT CTC counts and Δ% CTC (the percentage difference between pre-CRT and post-NCRT CTC counts) were independently associated with tumor response to NCRT. The Δ%CTC was also significantly higher in patients who achieved pCR. Recently, another study analyzed two specific markers in CTCs to evaluate pCR in LARC patients who underwent NCRT [91]. These two markers, thymidylate synthase (TYMS) and excision repair protein RAD23 homolog B (RAD23B), are thought to confer resistance to chemotherapy/radiotherapy. Therefore, CTCs were isolated and counted by ISET^®^ and then analyzed for the expression of TYMS/RAD23B to evaluate their predictive value. Patients who achieved pCR had lower CTC counts between the pre-NCRT and post-NCRT time points. In addition, none of the patients who achieved pCR expressed TYMS in all CTCs, while RAD23B expression on CTCs was detected in 100% of the nonresponders after NCRT. These findings show that dynamic monitoring of the presence of CTCs, CTC count, and CTCs expressing specific markers has utility in predicting the tumor response of LARC patients after NCRT and might prevent unnecessary radical surgery in good responders.

#### 3.2.2. Predicting Recurrence in Rectal Cancer Patients Who Undergo Neoadjuvant Chemo-Radiotherapy

Although the abovementioned CTC studies enrolled rectal cancer patients, only a small portion of patients received neoadjuvant treatment since surgical resection was always performed first in the early period. More recently, few studies have focused on rectal cancer as a separate entity for predicting recurrence (Table 3). For instance, a single-center trial involving 162 rectal cancer patients who received only preoperative short-course radiotherapy revealed that CTC detection 7 days after surgery was associated with local recurrence [92], though other studies have shown negative results. Using CellSearch^®^, Magni et al. [93] studied 85 LARC patients treated with NCRT, but only 18.8% of patients were positive for CTCs (defined by CTC ≥ 1) at baseline and 8.9% at 7 days after surgery, and no association was found between CTC count DFS. Similar results were found in another study, namely CK 20-positive CTCs by RT-PCR, with a 30% overall CTC detection rate. The presence of CTCs still failed to predict OS [94]. The authors stated that this was probably because NCRT leads to the clearance of CTCs in the blood, which is associated with a lower detection rate of CTCs after neoadjuvant treatment and surgical resection [95]. The relatively low CTC detection rate may be attributed to the use of older detection methods. Using a more advanced CTC detection technology (ISET^®^, Boulder, CO, USA), a size-based filter system to selectively isolate CTCs [96]), Silva et al. studied 63 LARC patients who underwent NCRT over a median follow-up time of 32 months. CTCs were detected in 88.9% of patients before NCRT, 71.9% after NCRT, and 54.3% after surgery. Multivariable regression for survival analysis revealed that a post-NCRT CTC count greater than the pre-NCRT CTC count was an independent predictor of DFS and OS [97]. Another pilot study conducted by Liu et al. used image flow cytometry (imFC) to detect peripheral CTCs (defined as CD45-/oHSV1-hTERT-GFP+/EpCAM+ cells) to assess the risk of recurrence in middle to lower LARC in a neoadjuvant setting with a median follow-up time of 49 months [98]. CTCs were detected before and after radiotherapy and during and after neoadjuvant chemotherapy. CTCs were detected in 76 of 83 patients (91.6%) at baseline, while a total of 74 patients underwent a CTC examination after radiotherapy; of these, CTCs were detected in 73 patients (98.8%). A ΔCTC count < −1 (postradiotherapy CTC count minus baseline count) was associated with pCR and persistent clinical complete response (cCR). The authors found that a CTC count > 3 was a high-risk factor for recurrence. Compared with low-risk patients, high-risk patients are the only independent factor for predicting recurrence. Survival analysis revealed that patients with a CTC count > 3 had significantly poorer 3-year RFS than patients with a CTC count ≤ 3. This study indicated the potential role of dynamic detection of CTCs in recurrence risk prediction and postradiotherapy decision-making in rectal cancer patients who undergo NCRT. In brief, the above studies show that as CTC detection techniques advance, there is potential for CTCs to become a prognostic factor in rectal cancer patients who undergo NCRT.

## 4. Recent Progress in Advanced CTC Detection and CRC Recurrence

Traditionally, the markers for CTC identification only include conventionally defined CTCs, such as epithelial cell adhesion molecule (EpCAM) [66]. However, several studies using the CellSearch^®^ system, the first and only FDA-approved device, showed a low CTC detection rate in nonmetastatic CRC patients [99] and therefore failed to predict CRC recurrence [6,100]. This limitation might be due to its EpCAM-dependent CTC isolation technology [101], which could pose a challenge for detecting other types of CTCs without EpCAM markers [102]. Therefore, with advances in CTC-detection techniques, other subtypes of CTCs can be investigated and utilized for detecting CTC recurrence.

### 4.1. CTC Subtypes and the Prognostic Value of CTC Subtypes in CRC Recurrence

Apart from the expression of EpCAM markers on CTCs, the heterogeneity of CTC subtypes has received attention as multiple advanced detection tools are being developed. For instance, the epithelial–mesenchymal transition (EMT) is a critical step in cancer metastasis and affects the phenotypes of CTCs [103]. During this process, epithelial CTCs (E-CTCs) are transferred into mixed CTCs (E/M-CTCs) and then converted into mesenchymal-phenotype CTCs (M-CTCs) to promote their invasive and metastatic ability [104]. Additionally, circulating cancer stem cells (CCSCs), which are capable of self-renewal, are responsible for initiating metastases and are associated with poor prognosis [105,106]. CTC clusters, another atypical CTC subtype, arise when cancer cells gather alone or form clumps with stromal or immune cells [107], and their presentation is strongly associated with a poor prognosis [108,109]. In this section, we discuss the current research using CTC subtypes (i.e., CCSCs, EMT-related CTCs, and CTC clusters) to predict postoperative CRC recurrence (Table 4).

#### 4.1.1. Epithelial–Mesenchymal Transition (EMT)-Related CTCs

EMT is a dynamic process in which epithelial cells convert into a mesenchymal phenotype to promote invasion and metastasis [103]. It is believed that the ability of cancer cells to maintain EMT phenotypic plasticity may drive cancer recurrence [110]. Earlier studies universally used epithelial-specific markers such as EpCAM and cytokeratins to recognize CTCs. Therefore, these methods perform poorly at detecting cells that undergo epithelial–mesenchymal transition or exhibit mesenchymal phenotypes [111]. Like one kind of CTC characterized by the suppression of epithelial markers (e.g., E-cadherin) and upregulation of mesenchymal markers, such as vimentin and N-cadherin, CTCs exhibit mesenchymal features (M-CTCs) and have been reported to be a significant indicator of poor prognosis in CRC patients [112]. Hence, identifying these subtypes of CTCs may help with prognostic risk evaluation. For instance, Yokobori and colleagues used mesenchymal-correlated CTCs (expression of Plastin 3, a marker that is overexpressed in the mesenchymal status of CTCs in CRC [113]) and its clinical association with the survival of CRC patients [114]. This large cohort included 711 stage I–IV CRC patients, 25% of whom presented with mesenchymal-correlated CTCs. Survival analysis revealed that patients with mesenchymal-correlated CTCs had significantly worse survival than patients with negative CTCs at all stages. According to the subgroup analysis, mesenchymal-correlated CTCs were an independent prognostic factor for disease recurrence in stage II and III CRC patients. Moreover, CRC patients with many M-CTCs present in the blood suffer from significantly worse overall survival than patients with low M-CTC counts [115]. In addition to examining the prognostic value of M-CTCs at a single time point, the concept of dynamic monitoring, which has been mentioned before, may provide more information for cancer monitoring. Shi et al. analyzed the kinetic changes in M-CTCs in stage II and III CRC patients [116]. The authors used an ISET-CTCBIOPSY^®^ device with an immunocytochemistry assay [117] to detect M-CTCs before surgery and after adjuvant chemotherapy. Both RFS and OS were significantly worse in patients whose M-CTCs were positive at both time points than in those whose M-CTCs were negative at the other two time points. Persistent M-CTC positivity before and after chemotherapy was an independent prognostic factor for shorter RFS and OS.

Taken together, these findings support that the presence of CTCs undergoing the EMT process and M-CTCs detected in the blood indicate a higher risk of CRC recurrence. The presence of M-CTCs after standard surgery and adjuvant chemotherapy implies that residual tumor cells are present, in which case intense surveillance programs or continuous aggressive systemic therapy should be considered.

#### 4.1.2. Circulating Cancer Stem Cells

A small subpopulation of CRC cells (~2.5%) in primary tumor tissue [118] exhibit two important abilities, namely self-renewal and efficient regeneration of the phenotypic heterogeneity of a parental tumor, and are responsible for initiating overt metastases [105,106]. On the basis of these properties, these types of cells are called circulating cancer stem cells (CCSCs), and their presentation is associated with poor tumor differentiation, advanced disease stage, and lymph node metastasis in CRC patients [119]. Therefore, the presence of CCSCs in the preoperative or postoperative blood samples of CRC patients may indicate poor prognosis. For instance, Katoh et al. examined the relationship between CD44v9-expressing CTCs and cancer prognosis in 150 stage I–IV CRC patients [120]. CD44v9 is believed to be an exceptionally important factor for cancer stem cells in CRC [121]. First, the proportion of CCSCs increased with increasing tumor stage, from 28.8% in the early-stage cohort to 36.4% in the stage III cohort and 63.9% in the stage IV cohort. Subgroup survival analysis of stage III patients revealed that the 5-year survival rate was 89.5% for patients with negative CCSCs and 52.4% for patients with positive CCSCs. Using flow cytometry, Lieto et al. demonstrated that CD26(+)/EpCAM(−) circulating cancer cells were prognostic risk factors for CRC recurrence [122]. CD26-positive cells have strong tumor invasion and chemoresistance abilities, and their presence in primary tumors predicts distant metastasis in CRC patients; thus, CD26-positive cells are believed to be one type of CCSC [123]. The authors also found that high levels of pre-operative CD26(+) cells predicted tumor recurrence (44.4% of patients with high CD26(+) cell counts had recurrence, while only 5% of patients with normal CD26(+) cell counts had recurrence). These results indicate that the identification of specific CCSCs may have clinical utility in risk assessment for CRC recurrence.

#### 4.1.3. Circulating Tumor Cell Clusters

CTC clusters, also known as circulating tumor microemboli (CTMs), are the phenomenon of CTCs gathering alone (homotypic CTC clusters) or CTC clumps with stromal or immune cells (heterotypic CTC clusters) [107]. Both homotypic and heterotypic CTC clusters showed a high metastasis-forming capability and high proliferation rate compared to single CTCs, and their presence in the peripheral circulation of cancer patients was associated with a poor prognosis [108,109]. For instance, a study involving 133 CRC patients [124] revealed that the presence of CTC clusters (homotypic CTC clusters) was more strongly correlated with metastatic disease, as the percentage of CTC clusters detected increased as the number of disease stages increased; from stage I to stage II to III to IV, the percentages were 6.8%, 15.4%, 20.8%, and 50%, respectively. Patients with CTC clusters exhibited significantly worse survival than cluster-free patients. Moreover, a larger study using a self-assembled cell array (SACA) chip system [125] to detect CTC clusters (homotypic CTC clusters) in 290 CRC patients produced promising results [126]. Subgroup analysis of stage III CRC patients showed that RFS was worse in patients with CTC clusters than in those with negative results, and CTC clusters were a risk factor for recurrence. On the other hand, Xu et al. published a retrospective study analyzing CTC and white blood cell (heterotypic CTC clusters) clusters in 329 CRC patients who underwent curative intent surgery [127]. CTCs were enriched and distinguished by the CanPatrol system, and 19% of patients were CTC–WBC cluster positive. With a median follow-up of 30 months, the authors reported that progression-free survival (PFS) was worse in patients with cluster-positive disease than in patients with cluster-negative disease. Their multivariate model further suggested that the CTC-WBC cluster was an independent risk factor for PFS. As a result, the presence of CTC clusters might be a strong prognostic factor for CRC recurrence.

In summary, CTC subtypes (e.g., EMT-related CTCs, CCSCs, and CTC clusters), which cannot be detected by conventional methods, have proven to be prognostic factors for disease recurrence or progression in CRC patients. The presence of these atypical CTCs is highly correlated with late-stage disease and may suggest that more aggressive surveillance and advanced therapy would benefit such patients.

**Table 4 cancers-16-02316-t004:** Prognostic value of CTC subtypes in postoperative CRC patients.

Study	CTC Subtypes	Technique	Blood Sample Timing	Definition of Subtype CTC Positive	Results
Yokobori et al., 2013 [114]	M-correlated CTC	RT-PCR	Before surgery	CTC presentation	The OS was significantly inferior for CTC (+) groups than for CTC (−) groups. CTC (+) was an independent prognostic factor for cancer recurrence in stage II and III patients.
Chen et al., 2022 [115]	M-CTC	CanPatrol	Multiple times	M-CTCs ≥ 5	OS was significantly inferior for M-CTC (+) groups than for M-CTC (−).
Shi et al., 2020 [116]	M-CTC	ISET CTCBIOPSY^®^	Multiple times	M-CTCs ≥ 1	Pre-operative M-CTC (+) group had significantly inferior RFS and OS than M-CTC (−) group. Post-adjuvant therapy M-CTC (+) group had significantly inferior RFS and OS than M-CTC (−) group. Persistent M-CTC (+) before and after anticancer treatment was an independent prognostic factor for shorter RFS and OS.
Katoh et al., 2015 [120]	CCSC	RT-PCR	N/A	CSC presentation	Recurrence rate was 40% for CSC (+) groups vs. 14.3% for CSC (−) groups in stage III CRC. The 5-year survival rate was 52.4% for CSC (+) groups vs. 89.5% for CSC (−) groups in stage III CRC.
Lieto et al., 2015 [122]	CCSC	Flow cytometry	Multiple times	CSC ≥ 1	Pre-operative and post-operative CSC (+) were both independent prognostic factors for cancer recurrence.
Chu et al., 2021 [124]	CTC-Clusters	SACA	N/A	Clusters presentation	PFS was significantly inferior for (+) cluster groups than for (−) clusters.
Hao et al., 2024 [126]	CTC-Clusters	SACA	During surgery	Clusters presentation	RFS was significantly inferior for (+) cluster groups than for (−) clusters in stage III patients. Clusters was an independent factor for RFS in stage III patients.
Xu et al., 2022 [127]	CTC-WBC Clusters	CanPatrol	Before surgery	Clusters presentation	PFS was significantly inferior for (+) cluster groups than for (−) clusters. Cluster was an independent factor for PFS.

M-correlated CTC = mesenchymal-correlated CTC; M-CTC = mesenchymal types of CTC; CCSC = circulating cancer stem cell; WBC = white blood cell; RT-PCR = reverse transcription polymerase chain reaction; ISET CTCBIOPSY^®^ = one- step isolation by size of epithelial tumor cells; SACA = self-assembled cell array; OS = overall survival; PFS = progression-free survival; RFS = recurrence-free survival. N/A = not applicable.

### 4.2. Combination of other Clinicopathological Markers and CTC-Related Factors to Predict CRC Recurrence

The abovementioned studies have revealed the potential of applying CTC and CTC subtypes as markers to predict CRC recurrence. However, their utility is not commonly exploited in clinical practice. The reason might be the rarity and heterogeneity of CTCs in blood samples, and the use of CTCs alone as a single marker might lead to misleading results. One review article summarized and discussed the power of combining CTC counts with other clinical parameters (e.g., CTC, CTC subtypes, CEA, carbohydrate antigen-199 (CA-199)) in predicting CRC recurrence to improve the accuracy of cancer detection, prognostic evaluation, and therapeutic response monitoring in numerous malignancies, including CRC [128]. This section will discuss the value of CTC detection combined with different clinical parameters to increase the predictive power and risk evaluation performance for CRC recurrence (Figure 2).

#### 4.2.1. CTCs Combined with Multiple Markers for the Detection of CRC Recurrence

Several aforementioned studies using CTCs to predict CRC recurrence have demonstrated that some clinicopathological factors were risk factors for CRC recurrence, such as tumor staging [58,59,63], lymph node metastasis [116,129], and perineural invasion [73,74]. By combining CTCs with other clinicopathological parameters, the coverage of detecting CRC recurrence could become wider and therefore could lead to the performance of advanced radiologic examinations (e.g., PET) and applying aggressive therapy earlier in high-risk patients.

Advanced analyses combining multiple markers have helped authors establish recurrence detection models for CRC patients after radical surgery (Table 5). For instance, Allen-Mersh et al. revealed that the combination of lymph node metastasis and the presence of CTCs 24 h after surgery was a stronger predictor of recurrence after CRC excision than single factors alone [57]. Uen et al. retrospectively analyzed two similar studies that combined CTCs and clinicopathological factors to better predict CRC recurrence. Factors including tumor invasion, vascular invasion, and the presence of CTCs in one study of a stage II CRC cohort [79] and lymph node metastasis, vascular invasion, and the persistent presence of CTCs in another stage I to III CRC cohort [72] were combined to assess recurrence risk. Remarkably, both of these studies illustrated a powerful ability to predict CRC recurrence compared with studies without any one risk predictor. Wang et al. established a prognostic model by Cox proportional-hazards regression analysis [69]. First, they found that tumor stage (III vs. II), postoperative carbohydrate antigen 72-4 (CA72-4) positive (a prognostic tumor marker of CRC recurrence [130]), and postoperative CTC count were independent predictors of cancer relapse in their cohort. They then assigned weighted scores of 3, 4, and 5 to positive post-CTC, stage III disease, and positive post-CA72-4, and patients were divided into different groups depending on their risk scores. According to the results, the predictive model demonstrated high sensitivity, specificity, and accuracy compared to postoperative CTCs alone for predicting CRC recurrence. Thus, the model combining CTCs and multiple other factors provided more extensive coverage for predicting CRC recurrence and might be used to identify patients at high risk of recurrence and guide aggressive treatment to improve their clinical outcomes.

#### 4.2.2. CTCs Combined with Multiple Markers for the Risk Evaluation of CRC Recurrence

CTCs combined with multiple markers not only increase the coverage of CRC recurrence detection but can also be applied to evaluate recurrence risk because patients with multiple risk factors have greater chances of CRC recurrence (Table 6). In a recently published study, CTC-related factors and the CEA level were combined to improve the predictive power for CRC recurrence [124]. The authors found that a CTC count > 3, CEA > 5 ng/ml, and the presence of CTC clusters were all individually associated with recurrence in stage III CRC patients (odds ratios of 2.6, 5.1, and 3.2, respectively). However, patients who are affected by all three factors are more likely to experience CRC recurrence (odds ratio: 17.1). A later study showed that combined parameters obtained before treatment and during surgery, including CTCs, CA19-9, and CTC clusters, in stage III CRC patients are associated with a greater risk of recurrence and can successfully predict early colorectal cancer recurrence after medical treatment [126].

Briefly, the application of multiple clinicopathological parameters (e.g., lymph node metastasis, vascular invasion), laboratory markers (e.g., CEA, CA-199, and CA-72-4), CTCs and CTC subtypes could be synthesized into a valuable prognostic risk model for predicting CRC recurrence and provide more information to physicians and patients to choose more appropriate, individualized surveillance strategies and possibly more aggressive medical treatment.

## 5. Discussion

Clinically, detecting CRC recurrence or assessing recurrence risk are important for applying immediate and precise interventions, such as systemic chemotherapy or another surgical resection, to improve patient survival. Although many surveillance methods (e.g., CEA, CT, and PET) have been used in clinical practice, 30−50% of CRC patients experience cancer recurrence and die from their cancer, indicating that there is an urgent need for new markers for detecting CRC recurrence. CTCs are a prognostic marker for CRC recurrence. However, because of the limitations of CTC detection markers (e.g., EpCAM) and lower CTC detection sensitivity via conventional methods, the role of CTCs as therapeutic markers for adjuvant chemotherapy in stage II CRC and their utility for predicting cancer recurrence in LARC patients receiving NCRT remain controversial. Nevertheless, as CTC detection technology has advanced (e.g., SE-iFISH and image flow cytometry), the CTC detection rate has increased, making CTC positivity a potential marker for deciding whether to apply adjuvant chemotherapy in patients with stage II CRC and overcoming the detection limitation for rectal cancer patients who undergo NCRT. Furthermore, to enhance the ability of CTCs to predict CRC recurrence, not only the application of CTC subtypes (e.g., EMT-related CTCs, CCSCs, and CTC clusters) but also the combination of CTC-related parameters with clinicopathological factors (e.g., tumor severity, lymph node metastasis, and vascular invasion) and laboratory markers (e.g., CEA, CA72-4, and CA-199) to establish a multimarker model for CRC recurrence are feasible strategies that will improve the detection rate and prediction performance for CRC recurrence. Briefly, studies have not only precisely predicted CRC recurrence but have also been used to guide clinical treatment. Many studies have demonstrated that CTC subtypes and CTCs combined with traditional clinicopathological parameters present greater predictive power for CRC recurrence than a single CTC factor. Nevertheless, studies exploring the role of CTCs combined with CTC subtypes and multimarker models in assisting with adjuvant chemotherapy or monitoring LARC recurrence after neoadjuvant therapy are still lacking. The application of this new strategy in clinical practice is worthy of further study. 

In addition to CTCs, circulating tumor DNA (ctDNA), another “liquid biopsy” marker, consists of small DNA fragments that are shed by the tumor into the bloodstream and has attracted increased amounts of attention in CRC research [131]. In contrast to CTCs, which reflect more metastases-initiating cells, ctDNA provides information on mutations and genomic changes throughout the entire tumor genome, which is highly correlated with cancer severity [132] and has been reported to be a potential prognostic marker for the early detection of CRC recurrence in an earlier study [133]. Moreover, one recently published meta-analysis showed that, compared with other surveillance tools (i.e., CEA, CT, and PET), ctDNA was most often positively correlated with CRC recurrence [46], indicating that the use of ctDNA may serve as a precise tool for monitoring cancer relapse during CRC follow-up. Moreover, ctDNA has been proven to guide adjuvant chemotherapy in stage II colon cancer patients [134]. In this pilot study, the 3-year RFS was 86.4% among ctDNA-positive patients who received adjuvant chemotherapy and 92.5% among ctDNA-negative patients who did not, indicating that ctDNA could be a marker of decision-making for applying adjuvant chemotherapy in stage II colon cancer patients. Based on the ability of ctDNA to detect CRC recurrence, combining CTC-related parameters and ctDNA might provide more accurate individual information for guiding adjuvant chemotherapy in stage II patients or even in stage III patients with a relatively low recurrence risk to avoid unnecessary chemotherapy treatment. Moreover, the genetic data obtained from ctDNA can provide variable information on drug sensitivity, drug resistance, and monitoring treatment response to effectively provide appropriate treatment strategies for CRC patients. 

Although CTCs are a potential prognostic factor for CRC recurrence, the utility of CTCs in rectal cancer patients who undergo neoadjuvant chemoradiotherapy remains controversial for several reasons. First, studies have only examined CTCs after surgical resection, and patients might still have minimal residual disease after surgery [48]. However, in the NCRT setting, systemic chemotherapy is given to eradicate micrometastatic disease earlier to improve disease-free survival [22]. Because NCRT leads to the clearance of CTCs in the blood, which is associated with a lower detection rate of CTCs after neoadjuvant treatment, it would be more challenging to detect CTCs if patients underwent systemic chemotherapy before surgery. Second, since dynamic monitoring of CTCs after every treatment (e.g., surgery, radiotherapy, and chemotherapy) is critical for predicting recurrence, multiple check points can provide more detail than a single test, while the actual counts of CTCs can also provide more information than can the mere presence or absence of CTCs. Few studies have applied CTC subtypes and multimarker models for predicting cancer recurrence in advanced rectal cancer patients who undergo NCRT, so well-designed studies are warranted. An alternative therapeutic strategy to NCRT, named total neoadjuvant treatment (TNT), for treating advanced rectal cancer is gaining popularity [135]. TNT delivers both systemic chemotherapy and neoadjuvant chemoradiotherapy before surgery and has been associated with significantly better pCR, DFS, and OS than NCRT [136], suggesting that TNT is a promising strategy in locally advanced rectal cancer [137]. Nevertheless, patients who undergo TNT must go through a longer period of systemic treatment before surgery, indicating that dynamic monitoring of the therapeutic response to guide the treatment strategy plays an important role in this setting. CTCs and CTC subtypes, as real-time markers that can be examined often, might have the potential to assist in deciding the best therapeutic strategies for these specific patients in clinical practice.

## 6. Conclusions

The ability to predict or detect CRC recurrence early enables physicians to properly apply aggressive treatment plans to maximize patient survival. Current surveillance tools have certain limitations in terms of recurrence detection, as 30−50% of CRC patients experience cancer recurrence after radical surgery. To address this issue, a novel marker(s) for CRC recurrence is needed. CTCs, a population of cancer cells that detach from the primary tumor and enter the bloodstream, have the potential to become a marker that is not only useful for predicting CRC relapse but also can guide the choice of treatment plans for stage II CRC. With recent advances, the development of techniques with higher CTC detection rates (e.g., image flow cytometry) has made the use of CTCs for monitoring cancer recurrence in advanced rectal cancer patients who undergo NCRT promising. Moreover, CTC subtypes (e.g., EMT-related CTCs, CCSCs, and CTC clusters), which cannot be detected by conventional EpCAM-dependent CTC methods, have proven to be prognostic factors for disease recurrence or progression in CRC patients. The presence of these atypical CTCs is highly correlated with late-stage disease and may suggest more advanced therapy. Importantly, a feasible research direction is to combine the results of conventional CTC counts, dynamic CTC counts, and CTC-related parameters with clinicopathological factors to establish a multimarker model that can improve the detection rate and risk evaluation performance of CRC recurrence. CTCs, whether through their raw count, the proportions of CTC subtypes, or combinations of CTC-related parameters with clinicopathological factors, serve as a potential prognostic indicator of CRC recurrence, helping clinicians predict CRC recurrence, guiding adjuvant chemotherapy in early CRC patients, and serving as a prognostic factor for LARC patients who undergo NCRT.

## Figures and Tables

**Figure 1 cancers-16-02316-f001:**
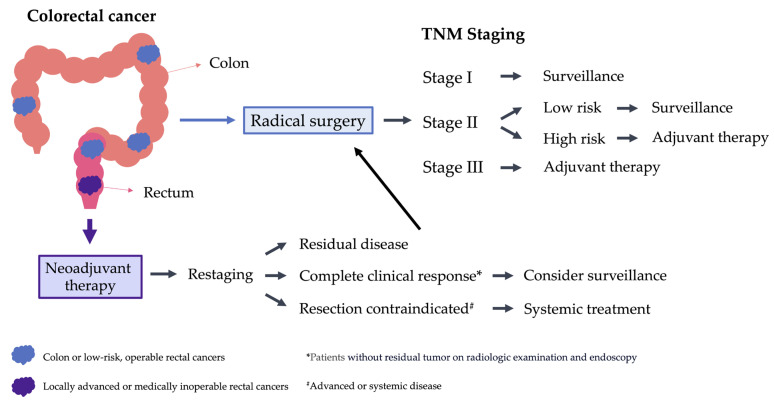
Treatment strategies of colorectal cancer without suspected or proven distant metastases. The blue arrow indicates the management of colon and operable rectal cancers, and the purple arrow shows the treatment of locally advanced or inoperable rectal cancers. The dark arrows show the process of suggesting strategies, depending on different cancer staging after surgery and response to neoadjuvant therapy.

**Figure 2 cancers-16-02316-f002:**
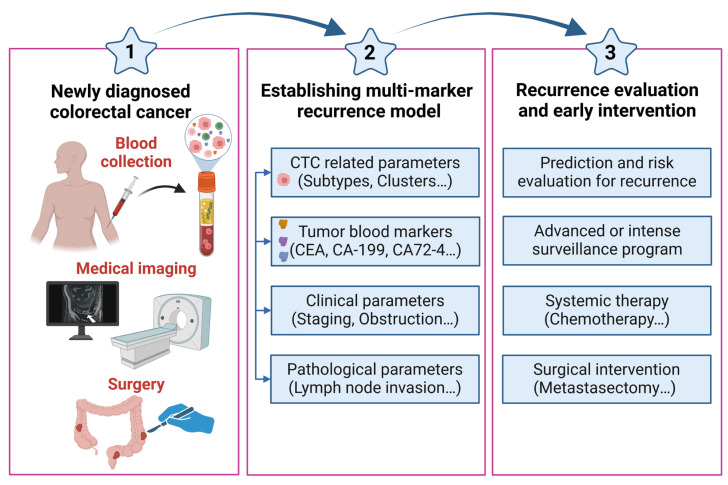
Brief description of the analysis of CTC-related parameters combined with multiple markers to establish model for CRC recurrence in clinical practice.

**Table 1 cancers-16-02316-t001:** Current surveillance tools suggested by NCCN guidelines.

Tools	Descriptions	Advantages	Limitations
Physical examinations	Evaluating objective anatomic findings through observation, palpation, percussion, and auscultation, e.g., digital examination	Costless Risk-free	Diagnosis is complicated and often delayed due to lack of specific symptoms
Carcinoembryonic antigen (CEA)	A specific blood tumor marker of CRC recurrence	Cost-effective Dynamic monitor Acceptable sensitivity	Insufficient sensitivity False positive due to many factors (e.g., smoking, inflammation, etc.)
Colonoscopy	An imaging tool using a fiber-optic flexible instrument inserted through the anus to examine the colon and rectum	Detect local recurrence Remove early adenomatous polyps Detect metachronous cancer	Bowel preparation before procedure Rare but severe risks during procedure (bleeding, perforation) Unable to detect extra-luminal recurrent disease
Computed tomography (CT)	An imaging tool combining X-rays and computer technology to produce images of the inside of the body.	Whole body survey Detect the presence of new lesions, especially in liver and lung.	Radiation exposure Low accuracy for discriminating postoperative change, extrahepatic metastases, and local recurrence.
Positron emission tomography (PET) scan	An imaging tool using radioactive drugs to evaluate the atypical metabolic activity of a particular organ or tissue	Whole body survey High sensitivity and specificity Highest accuracy for the detection of distant or local recurrence.	Radiation exposure Expensive Limited availability

**Table 2 cancers-16-02316-t002:** Clinical application of CTCs as a therapeutic-guiding marker for stage II CRC patients receiving adjuvant chemotherapy.

Study	Technique	Blood Sample Timing	Definition of CTC Positive	Results
Koch et al., 2006 [58]	RT-PCR	24 h after surgery	CTC presentation	CTC (+) is an independent prognostic factor for cancer recurrence.
Uen et al., 2007 [79]	Membrane-arrays	7 d after surgery	CTC presentation	CTC (+) is an independent prognostic factor for cancer recurrence. The RFS was inferior in CTC (+) groups vs. CTC (–) groups.
Yu et al., 2020 [80]	imFISH	7 d after surgery	CTCs ≥ 2	The RFS was 28.7 months for CTC (+) groups vs. 34.0 months for CTC (–) groups.
Yu et al., 2023 [81]	imFISH	7 d after surgery	CTCs ≥ 3	For patients who did not receive adjuvant chemotherapy, the OS was 68.1 months for CTC (+) groups vs. 82.0 months for CTC (–) groups.
Chen et al., 2022 [82]	SE-iFISH	7 d before surgery	CTCs ≥ 4	CTC (+) group had a significantly higher recurrence risk than CTC (–) group. CTC (+) group received greater benefits from adjuvant chemotherapy than CTC (–) group. If postoperative CTC (+) for more than 3 consecutive time points (2−6 months), the recurrence rate was 100%.

RT-PCR = reverse transcription polymerase chain reaction; imFISH = immuno-fluorescence in situ hybridization; SE-iFISH = subtraction enrichment with immunostaining-fluorescence in situ hybridization; RFS = recurrence-free survival; OS = overall survival.

**Table 3 cancers-16-02316-t003:** Prognostic value of CTCs in stage I–III rectal cancer patients who underwent neoadjuvant treatment followed by curative surgery.

Study	Neoadjuvant Setting	Technique	Blood Sample Timing	Definition of CTC Positive	Results
Nesteruk 2014 et al., [92]	RT: 25 Gy	RT-PCR	Multiple times	CTC presentation	CTC (+) at 7 days after surgery was of prognostic significance for the local recurrence.
Magni et al., 2014 [93]	RT: 45 Gy CT: capecitabine	CellSearch	Multiple times	CTC ≥ 1	No statistically significant association was found between CTC (+) and disease survival.
Hinz et al., 2015 [94]	RT: 50.4 Gy CT: 5-fluorouracil	RT-PCR	Before surgery	CTC presentation	No significant association was found between CTC (+) and OS.
Silva et al., 2021 [97]	RT: 50.4 Gy CT: 5-fluorouracil or capecitabine	ISET	Multiple times	CTC ≥ 1	Counts of post-NCRT CTC > pre-NCRT CTC was an independent predictor of DFS and OS.
Liu et al., 2023 [98]	RT: 25 Gy CT: XELOX or RT: 50 Gy CT: capecitabine	Image flow cytometry	Multiple times	High risk: CTC > 3 Low risk: CTC ≤ 3	CTCs counts <−1 (post-RT counts of CTC from baseline) is significantly associated with pCR and persistent cCR High-risk groups are the independent factor to predict recurrence when compared to low-risk patients Patients with CTCs > 3 have significantly poorer 3-year RFS than patients with CTCs ≤ 3.

RT = radiotherapy; Gy = gray; CT = chemotherapy; NCRT = neoadjuvant chemoradiotherapy; RT-PCR = reverse-transcription polymerase chain reaction; ISET = isolation by size of epithelial tumor cells; RFS = recurrence-free survival; OS = overall survival; pCR = pathological complete response; cCR = clinical complete response.

**Table 5 cancers-16-02316-t005:** Clinical utility of CTC combined multiple markers for recurrence detection in postoperative CRC patients.

Study	Combination	Parameter	Definition of Positive	Outcome
Allen-Mersh et al., 2006 [57]	CTC + Pathological parameter	CTCs	Presentation	Hazard ratio: 8.66
		LNs (+)	Presentation	Hazard ratio: 7.92
		CTCs or LNs (+) or both	Presentation	Hazard ratio: 18.54
Uen et al., 2007 [79]	CTC + Pathological parameter	Tumor invasion	Presentation	Hazard ratio: 4.08
		Vascular invasion	Presentation	Hazard ratio: 3.51
		CTCs	Presentation	Hazard ratio: 38.59
		At least one factor described above vs. no factors	Presentation	Hazard ratio: 27.12
Uen et al., 2008 [72]	CTC + Pathological parameter	LNs metastasis	Presentation	Hazard ratio: 7.65
		Vascular invasion	Presentation	Hazard ratio: 4.36
		CTCs	Presentation	Hazard ratio: 29.48
		At least one factor described above vs. no factors	Presentation	Hazard ratio: 7.06
Wang et al., 2019 [69]	CTC + Pathological parameter + Tumor markers	CTCs	Presentation	Sensitivity: 61.11%; Specificity 73.53%; AUC: 0.673
		TNM + CA72-4 + CTCs	Presentation	Sensitivity: 61.11%; Specificity 83.33%; AUC: 0.722
		CA72-4 + CTCs	Presentation	Sensitivity: 77.78%; Specificity 70.59%; AUC: 0.742

LN = lymph node metastasis; AUC = area under curve.

**Table 6 cancers-16-02316-t006:** Clinical utility of CTC combined multiple markers for recurrence risk evaluation in postoperative CRC patients.

Study	Combination	Parameter	Definition of Positive	Outcome
Chu et al. 2021 [124]	CTC + Tumor markers + CTC subtypes	CTCs CEA CA-19-9 Clusters CEA + clusters CA-19-9 + clusters CTCs + CEA CTCs + clusters CTCs + CEA + Clusters	>3 >5 >37 Presentation	Odds ratio: 2.6 Odds ratio: 5.1 Odds ratio: 6.2 Odds ratio: 3.2 Odds ratio: 4.8 Odds ratio: 5.4 Odds ratio: 7.3 Odds ratio: 8.4 Odds ratio: 17.1
Hao et al. 2024 [126]	CTC + Tumor markers + CTC subtypes	CTCs CEA CA-19-9 Clusters CA-19-9 + Clusters CTCs + CA19-9 + clusters CTCs + clusters	>1.5 >5 >37 Presentation	Odds ratio: N/A Odds ratio: 1.0 Odds ratio: 3.5 Odds ratio: 19.3 Odds ratio: 24.4 Odds ratio: 24.4 Odds ratio: 26.2

N/A = not applicable.

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
