# Peer review of "Current Applications and Future Directions of Circulating Tumor Cells in Colorectal Cancer Recurrence"

_cancers, 2024, doi:10.3390/cancers16132316_

Round 1
Reviewer 1 Report
Comments and Suggestions for Authors
The review article effectively emphasizes the significant role of circulating tumor cells (CTCs) in managing colorectal cancer (CRC), particularly in predicting recurrence and informing treatment strategies. It thoroughly examines the potential of CTCs in guiding adjuvant chemotherapy decisions for stage II CRC patients, presenting evidence from various studies that demonstrate CTCs' ability to identify high-risk individuals who may benefit from additional treatment. The discussion is well-supported by multiple studies, enhancing its scientific validity. Although the paragraph is generally well-crafted, some sentences could be improved for greater clarity and brevity. Additionally, Figure 1 should be revised for graphical enhancement to ensure higher image quality. The Tables provided are satisfactory and effectively complement the text.
Author Response
General Comments:
The review article effectively emphasizes the significant role of circulating tumor cells (CTCs) in managing colorectal cancer (CRC), particularly in predicting recurrence and informing treatment strategies. It thoroughly examines the potential of CTCs in guiding adjuvant chemotherapy decisions for stage II CRC patients, presenting evidence from various studies that demonstrate CTCs' ability to identify high-risk individuals who may benefit from additional treatment. The discussion is well-supported by multiple studies, enhancing its scientific validity. Although the paragraph is generally well-crafted, some sentences could be improved for greater clarity and brevity. Additionally, Figure 1 should be revised for graphical enhancement to ensure higher image quality. The Tables provided are satisfactory and effectively complement the text.
Response:
We are grateful to the reviewers for their insightful comments on our review paper. We have revised the manuscript to make it in a more concise way by following the suggestions provided by the reviewers. We also improve graphical quality of figure 1.
(Please refer to the figures in the manuscript: page 3, lines 102-108 or below)
Figure 1. Treatment strategies of colorectal cancer without suspected or proven distant metastases. The blue arrow indicates the management of colon and operable rectal cancers and the purple arrow shows the treatment of locally advanced or inoperable rectal cancers. The dark arrows are the process of suggesting strategies depends on different cancer staging after surgery and their response to neoadjuvant therapy.
We hope this revised response meets your requirements. We appreciate your review, thank you.

Reviewer 2 Report
Comments and Suggestions for Authors
This review “Current Applications and Future Directions of Circulating Tumor Cells in Colorectal Cancer Recurrence” by Kun-Yu Tsai et al., is an extensive account of the current status using the circulating tumor cells (CTCs) as a predictive marker for colorectal cancer (CRC) recurrence. The authors have quite nicely reported observations from many studies, which have used CTCs as a single marker or in combination with multiple other markers to predict the recurrence of CRC. The review is well structured and written well with updated information and appropriate citations. The authors aptly mentioned that despite significant progress in detection and the clinical procedure of surgical resection, nearly 30-35% CRC patient suffer from cancer recurrence and die. Therefore, there is an urgent need for early detection of the CRC recurrence, which will help guide the therapeutic strategy needed to prevent the cancer spread. CTC is certainly a useful tool to identify CRC recurrence. However, to rely on a single marker to detect recurrence is risky and also the lack of sensitivity in current methods to detect CTCs early, is a significant hindrance to prevent CRC recurrence. The authors have highlighted the limitations in detecting CTCs in CRC patients before or after surgery as recent studies have reported their diverse nature in terms of marker expression. Therefore, although CTCs can provide important information about CRC recurrence, to get the information as early as possible, additional markers need to be employed to the detection method to make the outcome consistent and correct. Overall, this is a very informative review on the current progress in the field.
However, the review has some limitations, which need attention.
1. It is a bit too long and the necessary information could be provided in a more concise way to keep the reader’s attention.
2. Although the information presented are important, the review is text heavy. A few illustrations will attract the readers more to the issue at focus.
3. Not all abbreviations are expanded in full at the point of first use. The authors need to check the text carefully and rectify that.
4. A definitive conclusion regarding the use of CTCs and which additional markers to be used as a predictive method for CRC recurrence is still not there. This is particularly important as the early detection will help decide the future strategy about adjuvant chemotherapy.
Author Response
General Comments:
This review “Current Applications and Future Directions of Circulating Tumor Cells in Colorectal Cancer Recurrence” by Kun-Yu Tsai et al., is an extensive account of the current status using the circulating tumor cells (CTCs) as a predictive marker for colorectal cancer (CRC) recurrence. The authors have quite nicely reported observations from many studies, which have used CTCs as a single marker or in combination with multiple other markers to predict the recurrence of CRC. The review is well structured and written well with updated information and appropriate citations. The authors aptly mentioned that despite significant progress in detection and the clinical procedure of surgical resection, nearly 30-35% CRC patient suffer from cancer recurrence and die. Therefore, there is an urgent need for early detection of the CRC recurrence, which will help guide the therapeutic strategy needed to prevent the cancer spread. CTC is certainly a useful tool to identify CRC recurrence. However, to rely on a single marker to detect recurrence is risky and also the lack of sensitivity in current methods to detect CTCs early, is a significant hindrance to prevent CRC recurrence. The authors have highlighted the limitations in detecting CTCs in CRC patients before or after surgery as recent studies have reported their diverse nature in terms of marker expression. Therefore, although CTCs can provide important information about CRC recurrence, to get the information as early as possible, additional markers need to be employed to the detection method to make the outcome consistent and correct. Overall, this is a very informative review on the current progress in the field.
However, the review has some limitations, which need attention.
Response:
We are grateful to the reviewers for their insightful comments on our review paper. We have revised the manuscript to reflect all of the suggestions provided by the reviewers. We have highlighted the changes within the manuscript.
Here is a point-by-point response to the reviewers’ comments and concerns.
Specific comment 1:
It is a bit too long and the necessary information could be provided in a more concise way to keep the reader’s attention.
Response:
Thank the reviewer for your suggestion indeed. We have revised article as suggested.
- (Please refer to the manuscript of highlighted revision: page 2, lines 70-81 or below).
For these patients, multiple clinical and histopathological factors, including advanced tumor stage, tumor perforation, intestinal obstruction, poorly differentiated histology, lymphovascular invasion or perineural invasion, have been evaluated as risk factors for recurrence. For instance, study found that the 5-year disease free survival (DFS) rate was 87.3% for stage II CRC patients with only one risk factor, but it dropped to 74.8% for patients with two or more risk factors [16]. Gertler et al. also found that the 5-year recurrence-free survival (RFS) of patients with stage II colon cancer decreased from 91% with no risk factor to 75% with three risk factors[17]. Nevertheless, current evidence has not identified any of the regularly assessed parameters as individually powerful enough for risk stratification in high-risk stage II CRC[18]. Research to better identify risk factors for those patients who may benefit from adjuvant chemotherapy is needed.
- (Please refer to the manuscript of highlighted revision: page 7, lines 232-234 or below).
CTCs are rare cancer cells that detach from primary tumors or metastatic tumors then circulate in the bloodstream [51] and it is believed that when CTCs are trapped in the capillaries of organs or tissue [52], the cancer will colonize and metastasize [53].
- (Please refer to the manuscript of highlighted revision: page 13, lines 510-513 or below).
Using CellSearch® [93]. Magni et al studied 85 LARC patients treated with NCRT, but only 18.8% of patients were positive for CTCs (defined by CTC ≥ 1) at baseline and 8.9% at 7 days after surgery, and no association was found between CTC count DFS.
- (Please refer to the manuscript of highlighted revision: page 13, lines 519-522 or below).
Using a more advanced CTC detection technology (ISET®, a size-based filter system to selectively isolate CTCs [96]), Silva et al. studied 63 LARC patients who underwent NCRT over a median follow-up time of 32 months. CTCs were detected in 88.9% of patients before NCRT, 71.9% after NCRT and 54.3% after surgery.
- (Please refer to the manuscript of highlighted revision: page 28, lines 1098-1100 or below).
Reference [53]: Zhan, Q.; Liu, B.; Situ, X.; Luo, Y.; Fu, T.; Wang, Y.; Xie, Z.; Ren, L.; Zhu, Y.; He, W.; et al. New insights into the correlations between circulating tumor cells and target organ metastasis. Signal Transduct Target Ther 2023, 8, 465, doi:10.1038/s41392-023-01725-9.
Specific comment 2:
Although the information presented are important, the review is text heavy. A few illustrations will attract the readers more to the issue at focus.
Response:
Thank you for your suggestions, we have add an illustration of CTC related parameters combining clinicopathological factors to establish multimarkers model for CRC recurrence in clinical practice.
(Please refer to the figures in the manuscript: page 20, lines 756-759 or below)
Figure 2. A brief description of the analysis of CTC related parameters combined with multiple markers to establish model for CRC recurrence in clinical practice.
(Licensing right: Agreement number LW26YFWVBU)
Specific comment 3:
Not all abbreviations are expanded in full at the point of first use. The authors need to check the text carefully and rectify that.
Response:
Thank you for pointing this out. We have revised multiple parts to reflect your suggestion.
(Please refer to the manuscript of highlighted revision: page 3, lines 122 or below).
computed tomography (CT)
(Please refer to the manuscript of highlighted revision: page 7, lines 275 or below).
Food and Drug Administration (FDA)
(Please refer to the manuscript of highlighted revision: page 8, lines 289 or below).
ISET (isolation by size of epithelial tumor cells)
(Please refer to the manuscript of highlighted revision: page 8, lines 310-311 or below).
adjuvant mFOLFOX (oxaliplatin and 5-FU) chemotherapy
(Please refer to the manuscript of highlighted revision: page 12, lines 310-311 or below).
cytokeratin 20 (CK20)
(Please refer to the manuscript of highlighted revision: page 12, lines 495 and 497 or below).
NCRT
(Please refer to the manuscript of highlighted revision: page 14, lines 553or below).
Gy = gray.
(Please refer to the manuscript of highlighted revision: page 20, lines 747or below).
carbohydrate antigen-199 (CA-199)
(Please refer to the manuscript of highlighted revision: page 20, lines 780or below).
carbohydrate antigen 72-4 (CA72-4)
(Please refer to the manuscript of highlighted revision: page 25, lines 942or below).
(e.g., EMT-related CTCs, CCSCs and CTC clusters)
Specific comment 4:
A definitive conclusion regarding the use of CTCs and which additional markers to be used as a predictive method for CRC recurrence is still not there. This is particularly important as the early detection will help decide the future strategy about adjuvant chemotherapy.
Response:
Thank you for pointing this out. It would have been interesting to explore this aspect. Regarding the concept of CTC and other parameters for CRC recurrence assessment, current research is not enough to make a complete conclusion. More well-designed studies are needed in the future, but we still give our insights based on the existing data and identify potential candidates. We have added more information to emphasize this point.
(Please refer to the manuscript of highlighted revision: page 25, lines 878-884 or below).
Furthermore, to enhance the ability of CTCs to predict CRC recurrence, not only the application of CTC subtypes (e.g., EMT-related CTCs, CCSCs and CTC clusters) but also the combination of CTC-related parameters with clinicopathological factors (e.g., tumor severity, lymph node metastasis and vascular invasion) and laboratory markers (e.g., CEA, CA72-4 and CA-199) to establish a multimarker model for CRC recurrence are feasible strategies that will improve the detection rate and prediction performance for CRC recurrence.
(Please refer to the manuscript of highlighted revision: page 25, lines 888-891 or below).
Nevertheless, studies exploring the role of CTCs combined with CTC subtypes and multimarker models in assisting with adjuvant chemotherapy or monitoring LARC recurrence after neoadjuvant therapy are still lacking. The application of this new strategy in clinical practice is worthy of further study.
We hope this revised response meets your requirements. We appreciate your review, thank you.
